



# A model for the Artic mixed layer circulation under a melted lead: Implications on the near-surface temperature maximum formation

Alberto Alvarez

NATO-STO Center for Maritime Research and Experimentation-CMRE, V. San Bartolomeo 400, La Spezia, 19126, Italy

*Correspondence to*: Alberto Alvarez (alberto.alvarez@cmre.nato.int)

**Abstract.** Leads in the sea ice pack have been extensively studied due to their climate relevance. An intense heat exchange between the ocean and the atmosphere occurs at leads in winter. As a result, a major salt input to the Arctic mixed layer is generated at these locations by brine rejection. Leads also constitute preferential melting locations in the early melting season, but their oceanography and climate relevance, if any, still remain unexplored during this period of the year. This study

investigates the oceanographic circulation under a melted lead, resulting from the combined effect of the lead geometry, solar radiation and sea ice melting. Results derived from an idealized framework, suggest the daily generation of near surface convection cells that extend from the lead sides to the lead center. Convection cells disappear when melting is diminished during the period of minimum solar insolation. The cyclical generation and evolution of convection cells with the solar cycle, impacts the heat storage rate in the mixed layer below the lead. The contribution of this circulation pattern to the generation of

the Near Surface Temperature Maximum (NSTM), is discussed in terms of its capability to inject warm surface waters below the open and sea ice surface. It has been suggested that the NSTM probably affects the oceanographic structure and acoustic properties of the upper ocean and the overlying ice cover.

## 1 Introduction

The Arctic Ocean is currently responding to changes in atmospheric and oceanic processes associated to global warming, by a rapid retreating of sea ice (Liu et al., 2013; Stroeve et al., 2007). Ice retreating not only affects the local Arctic environment, but also has a feedback on the global climate. The increase of abnormal weather events in the Northern Hemisphere and the

modifications in the stability of the thermohaline circulation, are among the climate components influenced by ice melting (Levermann et al., 2007; Vihma, 2014).

Sea ice loss involves heat transfers in the ice-atmosphere-ocean system mediated by physical mechanisms like the atmospheric and oceanic heat transports (Zhang et al., 2008; Spielhagen et al., 2011), variations in water vapor and cloudiness (Schweiger et al., 2008), or modifications in the sea ice cover (Screen and Simmonds, 2010). In particular, cracks in the sea ice pack (also

known as leads) are ice-free areas where strong heat and chemical exchanges between the ocean and the atmosphere occur (Maykut, 1978; Alam and Curry, 1995; Douglas et al., 2005; Kort et al., 2012; Steiner et al., 2013). Leads are typically generated by stress-deformation events in the continuous sea ice pack, that develop recurring and elongated features of open water and thin ice. Their geometry ranges from 10 m to 1 km wide and up to 100 km long (Wilchinsky et al. 2015). Due to their climate relevance, a number of studies have considered the identification and characterization (width, orientation, area

coverage and spatial distribution) of leads in the ice cover (Barry et al., 1989; Miles and Barry, 1998; Brohan and Kaleschke, 2014; Wernecke and Kaleschke, 2015; Hoffman et al. 2019).

In winter, leads rapidly refreezes with frazil ice production due to the large heat fluxes from the exposed relatively warm ocean to the cold air. The winter heat exchange between the ocean and the atmosphere is so intense in leads, that the total heat flux





through them is equivalent to the heat flux through the continuous ice pack, although leads represent less than 1–2 % of the

total ice cover (Maykut, 1982; Perovich et al., 2011). Seawater refreezing in leads generates a buoyancy flux by brine rejection. This phenomenon results from the smaller capacity of sea ice to dissolve salt. While freezing in the leads, seawater releases much of its salt into the underlying water causing its densification and sinking. The process generates a particular circulation pattern at the lead location: the dense water flows away from the lead when reaching the bottom of the mixed layer. Conversely, freshwater flows in from the lead sides near the surface (Kozo, 1983; Morison et al, 1992). The salt rejected by brine rejection

constitutes a major salt input to the Arctic mixed layer (Morison et al, 1992).

As winter progress, the new ice produced in the leads is covered by drifting snow to reappear in early June when the snow melts. Leads constitute then preferential melting locations in the early melting season (Perovich et al., 2001). This is because the new ice in the leads is thinner and has a lower albedo than the adjacent ice (Grenfell and Maykut, 1977; Tschudi et al., 2002). In addition lead locations are topographically lower, collecting meltwater which further reduce their albedo. Perovich

et al. (2001) found that ice in the leads is almost completely melted by the end of July. The ice landscape is then constituted by large plate-like separated by long, narrow open leads at this time. As the thermal deterioration of the sea ice progresses, this landscape transforms into a complex mosaic of floes interlaced with open waters (Perovich et al., 2001).

Although the plate-like geometry of the ice landscape in mid-summer may resemble the winter one, the heat transfer between ocean and atmosphere in the lead is reversed, with ice melting instead of seawater freezing. This is expected to trigger a

circulation pattern in the open lead, different from that observed in winter. The theoretical/observational characterization of the mixed layer under a lead during this period, still remains an open issue.

A near surface temperature maximum (NSTM) has been observed in summertime at typically depths of 10-30 m in different Arctic Basins (Maykut and McPhee, 1995; Jackson et al., 2010; Carmack et al. 2015). In addition to the increased temperature, the low-salinity in the NSTM was associated to the sea ice melt (McPhee et al., 1998) or even river outflows (Macdonald et

al., 2002). Although the formation mechanism of the NSTM has not been fully established, observations demonstrate that this mechanism is seasonal (Kadko, 2000). For this reason, different authors have suggested that the NSTM results from the penetration of solar radiation through leads during summer (Maykut and McPhee, 1995; Jackson et al., 2010; Kadko, 2000). The heat contained in the NSTM remains trapped, when the accumulated fresh water from the sea ice melting separates the NSTM from the surface mixed layer. The strong near-surface stratification preserves the NSTM for long periods, probably

affecting the oceanographic structure and acoustic properties of the upper ocean and the overlying ice cover. In particular, sound ducts with a significant capability for long range acoustic propagation may result from the formation of the NSTM (Freitag et al., 2012).

This study numerically investigates the Arctic mixed layer circulation under a melted lead and its effect, if any, in the heat exchanges through the lead. The ultimate objective of the study is to assess if the circulation pattern under the lead could

contribute to the formation of the NSTM. The article is divided as follows: Section 2 details the theoretical and numerical framework. Results are presented in Section 3. Finally, Section 4 discusses and concludes the study.

## 2 Methodology

### 2.1 The physical model

The conceptual framework considers a vertical cross-section of an ice landscape described by recurring and large sea ice plates separated by long, narrow, rectilinear leads. The ice landscape is idealized by a stripe pattern with ice-patches and open water leads of 250m and 50m wide, respectively, Figure 1. Field observations reveal that maximum occurrence corresponds to leads and refrozen leads with less than 100m wide (Barry et al., 1989, Tschudi et al., 2002) and lead spacing below 500m (Haggerty et al., 2003). Thus, the selected geometry attempts to be representative of the main lead fraction. The ice plates are assumed


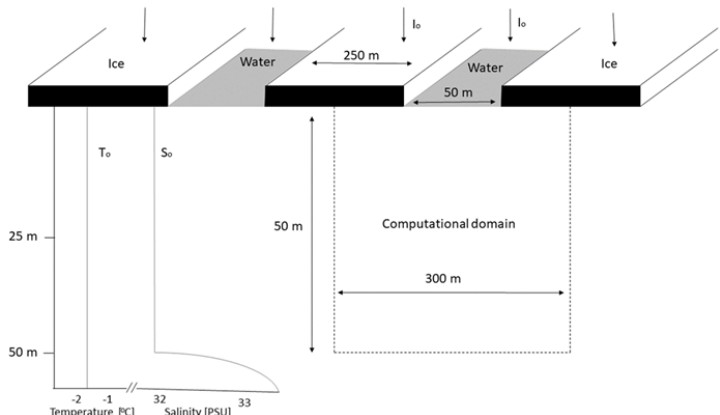

**Figure 1: Idealized framework of the current study. $I_o$, $T_o$ and $S_o$ refer to the incoming solar radiation and ground values of temperature and salinity, respectively.**

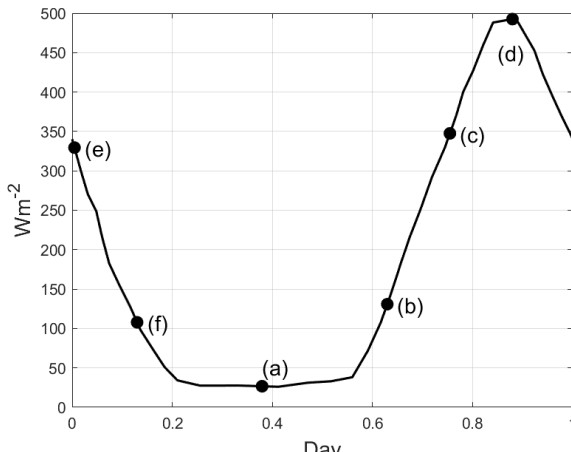


**Figure 2: Incoming shortwave radiation (adapted from McPhee, M. G., Stanton, T. P., 1996. Turbulence in the statically unstable oceanic boundary layer under arctic leads. J. Geophys. Res., 101, 6409-6428, with the permission of John Wiley and Sons, license number 4932450216592). Geographical location is north of the Prudhoe Bay (70°15′19″N 148°20′14″W). Maximum insolation is around 2130 UT. Alphabetic labels correspond to the states reported in the text.**


motionless. The mixed layer is bounded at the bottom by a pycnocline located at 50m depth, which is a characteristic depth for the pycnocline in the Eurasian Arctic Basin (Davis et al., 2016).

Solar radiation is the only energy source in the idealized scenario. The rationale of this assumption is discussed latter. The daily incoming shortwave radiation, $I_o$, was obtained from McPhee and Stanton (1996), Figure 2. The albedo for the bare ice

and open sea are 0.63 and 0.02, respectively (Bitz and Lipscomb, 1999; Li et al, 2006). The solar radiation is absorbed by the water body according to the Beer's law with an extinction coefficient λ of 0.08 m$^{-1}$ (Jackson et al., 2010). Below the ice plate, the fraction of solar radiation that penetrates the upper ice surface is assumed to be $i_o = 0.3[10 / (100.H_{ice}+10)]$ for an ice thickness ($H_{ice}$) of 1 m (Blitz and Lipscomb, 1999). Sea ice plates melt due to the incident solar radiation. The effects of melting on ice-plate geometry are not considered here.

A nonhydrostatic, Boussinesq, rotating, two-dimensional model is used in this study. The dynamical equations are given by:



$$\frac{\partial u}{\partial t} + u\frac{\partial u}{\partial x} + w\frac{\partial u}{\partial z} = -\frac{1}{\rho_0}\frac{\partial \delta p}{\partial x} + fv + \nu_T \nabla^2 u \tag{1}$$

$$\frac{\partial v}{\partial t} + u\frac{\partial v}{\partial x} + w\frac{\partial v}{\partial z} = -fu + \nu_T \nabla^2 v \tag{2}$$

$$\frac{\partial w}{\partial t} + u\frac{\partial w}{\partial x} + w\frac{\partial w}{\partial z} = -\frac{1}{\rho_0}\frac{\partial \delta p}{\partial z} - \frac{1}{\rho_0}\delta\rho g + \nu_T \nabla^2 w \tag{3}$$

$$\frac{\partial u}{\partial x} + \frac{\partial w}{\partial z} = 0 \tag{4}$$

For velocity components u, v, w and pressure ($\delta p$) and density ($\delta\rho$) anomalies. At near frozen temperatures the density variation is more dependent on salinity than temperature. Thus the simple equation of state $\rho = 0.808\ S + 1000$, where S is the salinity in g kg$^{-1}$ is considered (Smith and Morison, 1998). The Coriolis parameter is f=1.4 10$^{-4}$ s$^{-1}$ and g=9.8 ms$^{-2}$. $\nu_T = \nu + \nu_E$

where $\nu = 10^{-6}$ m$^2$/s is the molecular viscosity and $\nu_E$ is the eddy viscosity coefficient that accounts for the effects of the sub-grid scales. The parametrization of $\nu_E$ suggested by Smagorinsky (1963, 1993) is adopted here. This is one of the most popular subgrid scale models parameterizing eddy viscosity. Briefly, the Smagorinsky model assumes that $\nu_E$ is proportional to the absolute strain rate:

$$\nu_E = l_o^2 |S| \tag{6}$$

$$|S| = \sqrt{2 S_{ij} S_{ij}} \tag{7}$$

$$S_{ij} = \frac{1}{2}\left(\frac{\partial u_i}{\partial x_j} + \frac{\partial u_j}{\partial x_i}\right) \tag{8}$$

where $|S|$ is the absolute value of the strain rate tensor $S_{ij}$ and $u_i = \{u, v, w\}$ is *i-th* velocity component along the $x_i = \{x, y, z\}$

direction. The mixing length scale $l_o$ is parametrized using the Smagorinsky constant Cs, $l_o$=Cs Ls. $C_S$ has been tuned from 0.1 to 0.7 and it is set to 0.2 in the present study, which is a common default value. $L_S$ is a length scale representing a filtering width. Its value depends on the grid discretization. In two-dimensional flows, $L_S$ may be taken as the square root of the area of the computational element. Considering the symmetry of the present setting, the total viscosity $\nu_T$ is then given by:

$$\nu_T = \nu + \underbrace{(C_S\,L_S)^2\sqrt{\left(\frac{\partial u}{\partial x}\right)^2 + \left(\frac{\partial w}{\partial z}\right)^2 + 0.5\left(\frac{\partial u}{\partial z} + \frac{\partial w}{\partial x}\right)^2 + 0.5\left(\frac{\partial v}{\partial x}\right)^2 + 0.5\left(\frac{\partial v}{\partial z}\right)^2}}_{\nu_E} \tag{9}$$

Free-stress boundary conditions $\frac{\partial u}{\partial z} = 0, \frac{\partial v}{\partial z} = 0, w = 0$ are prescribed at the open water surface and the bottom pycnocline depth. Non-slip conditions (u=0, v=0, w=0) are specified at the sea ice plates. Finally, periodic boundary conditions are imposed at the lateral boundaries.

The temperature and salinity fields evolve according to the equations:

$$\frac{\partial \delta T}{\partial t} + u\frac{\partial \delta T}{\partial x} + w\frac{\partial \delta T}{\partial z} = K_T\,\nabla^2 \delta T + S_r - \frac{\partial \overline{T}}{\partial t} \tag{10}$$

$$\frac{\partial \delta S}{\partial t} + u\frac{\partial \delta S}{\partial x} + w\frac{\partial \delta S}{\partial z} = K_S\,\nabla^2 \delta S - \frac{\partial \overline{S}}{\partial t} \tag{11}$$

Where $\delta T(x, z, t)$ and $\delta S(x, z, t)$ are the temperature and salinity anomaly fields with respect to the reference fields T$_{ref}$ = T$_o$+$\overline{T}(t)$ and S$_{ref}$ = S$_o$+$\overline{S}(t)$, where $\overline{T}(t)$ and $\overline{S}(t)$ are spatial averages of the time varying component of the fields. Ground values are T$_o$ =-1.8 ºC and S$_o$ = 32.1 (Davies et al., 2016). According to Lei and Patterson (2002), $\frac{\partial \overline{T}}{\partial t} = \frac{H_o}{h}$ where H$_o$ represents





the heating intensity at the surface ($I_o/\rho_o C_p$ with $C_p$ the specific heat) and h is the depth of the domain. Similarly $\frac{\partial \overline{S}}{\partial t} = \frac{Q_s}{h}$, with $Q_s$ being the surface salinity flux at the top interface. A simple thermodynamic model is employed to compute the latter. Specifically, $Q_s$ is obtained from:

$$F = \frac{Q_m}{\rho_i L_f} \qquad (12)$$

$$Q_s = F(S_o - S_i) \qquad (13)$$

where F is the melting rate, $\rho_i$ is a characteristic sea ice density (900 kg/m³), $L_f$ is the latent heat of fusion of sea ice (333 kJ/kg) and $S_i$ (4 psu) is the the sea ice salinity content. $Q_m$ is the heat flux used in melting the bottom of the ice. This is computed using the linear relationship with the solar heat input to the ocean found by Perovich et al. (2011). The melting model (12) and

(13) ignores many aspect of the sea ice thermodynamics, but it is complex enough to capture the essential features of sea ice melting required in the present idealized framework. Sr is the internal heat source that quantifies the volumetric absorption of the solar radiation by the water body (Mao et al, 2010):

$$S_r = H_o \, \lambda \, e^{\lambda z} \qquad (14)$$


where $\lambda$ and $H_o$ have been previously defined. $K_{T,S}$ are defined by (Schumann, 1996):

$$K_{T,S} = k_{T,S} + \frac{v_E}{Pr_{sgs}} \qquad (15)$$

where $k_T = 10^{-7}$ m²/s ($k_S = 10^{-9}$ m²/s) is the molecular thermal (salinity) diffusion and $Pr_{sgs} = 0.4$ is the turbulent Prandtl number of the subgrid scale motions (Schumann, 1996).

The heat flux previously described defines the boundary condition for the temperature on the open sea surface. Instead, a Dirichlet condition is prescribed at the bottom of the ice plates ($\delta T(t) = -0.05°C - \overline{T}(t)$) and pycnocline depth ($\delta T(t) = -0.0°C - \overline{T}(t)$). A Neumman boundary condition, $\frac{\partial \delta S}{\partial n} = 0$ where n is the direction normal to the boundary, is selected for the salinity

field at the open sea surface and pycnocline depth. This boundary condition approximates null and negligible salt flux across the open sea interface and the pycnocline, respectively. Negative salt fluxes define the boundary conditions at the ice plates, representing the fresh water flux due to ice melting. Salinity fluxes across this boundary vary in time, depending on the solar heating. Similarly to the velocity field, periodic boundary conditions in $\delta T$ and $\delta S$ are prescribed at the lateral boundaries. Initial conditions are $\delta T(x, z, 0) = \overline{T}(0) = 0$ and $\delta S(x, z, 0) = \overline{S}(0) = 0$ in the model domain.


## 2.2 Computational approach

A standard Galerkin finite element method has been employed to spatially discretize Eqs. (1-4) and (6-7). The domain geometry was tessellated into 19604 triangular elements for this purpose. The characteristic sizes of the elements range from 0.35 m at the open sea surface to 1.75 m at the lateral boundaries, Figure 3. The discretization results in an algebraic system

of equations for the values of the fields at the corners of the triangular elements (nodes). Once solved, low-order piecewise





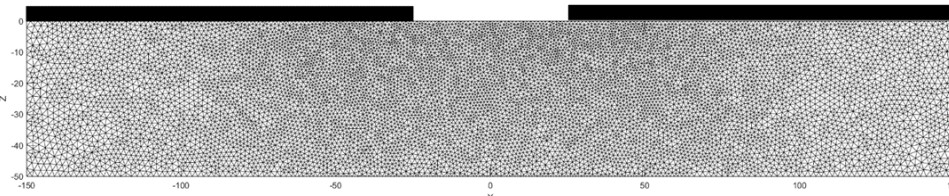


**Figure 3: Triangular meshing of the computational domain. Black rectangles represents the sea ice cover.**

polynomials are usually employed to interpolate the solution from the nodes to other locations of the physical domain. The specific mathematical expressions of the procedure are not replicated here, as they can be found in textbooks about finite

elements (e.g., Dhatt and Touzot, 1984; Zienkiewicz and Taylor, 1995).

A fractional step method is selected to integrate in time Eqs. (1-4). The approach is based on an operator splitting that yields a decoupling of the convection and diffusion of the velocity and the pressure, which acts to enforce the incompressibility constraint (Chorin, 1968). The time integration of the equations is performed by means of a Crank-Nicolson (implicit) time splitting for the convection and diffusion terms. An adaptive time step is chosen to ensure the stability of the numerical method.

The full model has been run for 15 simulation days.

**3 Results**

The particular analysis during days 12 and 13 follows as representative of the oceanographic conditions obtained after the model spin up. In particular, a clear mirror symmetry with respect to the lead center is found in the circulation pattern during

the hours of maximum solar insolation (489.5 $Wm^{-2}$). A northward (southward) flow of 0.005m/s develops at the eastern (western) part of the domain. Figure 4a shows the salinity distribution and current field obtained during this time period. Ice melting generates a layer of relatively fresh waters in the first 5 m depth, Figure 4a. A horizontal salinity gradient exists in this layer, being the water fresher under the ice caps than at the lead center. The horizontal salinity gradient induces near surface convection cells below the lead surface and the ice-plates. The velocity field converges at the geometric center of the open sea

surface and diverges at about 5 m depth. Convection cells extend laterally for more than 60 m with velocities of 0.015 m/s. This circulation pattern has a profound implication in the spatial distribution of the thermal field, Figure 4b. Specifically, surface currents flowing in from the lead sides accumulate heated waters at the lead center. At this location, warm waters are injected below the sea surface and distributed laterally by the convection cells, Figure 4b. Maximum temperature is -1.26 ºC. Convection cells weaken with the reduction of solar radiation, being unnoticeable when solar heating drops below 100 $Wm^{-2}$.

In particular, Figure 5a shows the salinity distribution obtained during the hours of minimum solar insolation (23.7 $Wm^{-2}$). The field is horizontally homogeneous and with a stable stratification in the vertical. Figure 5b displays the thermal field and the streamlines of the resulting circulation structure. The temperature distribution is described by a core of warm waters (> -1.5ºC) that extends from the surface up to 20 m depth in the water column. Temperature decreases laterally from the lead center. No coherent circulation patterns with maximum velocities of a couple millimeters per second, result during this period.

Figure 6 compares the sequence of events during a daily cycle, starting from the system state at the minimum insolation rate (23.7 $Wm^{-2}$), Figure 6a. Convection cells growth from the lead edges, with increasing solar heating. In particular, Figure 6b shows the circulation pattern obtained when solar heating reaches 130 $Wm^{-2}$. The cells are fully developed when solar radiation reaches 347 $Wm^{-2}$, Figure 6c. Interleaving of warm waters below the ice sheet is evident at this stage. The cells geometry when the solar insolation (489.5 $Wm^{-2}$) and melting are maximum, is display in Figure 6d. Cells weakens with the decline of solar

radiation. Figure 6e and Figure 6f display the evolution until reaching the state of minimum solar forcing, Figure 6a.



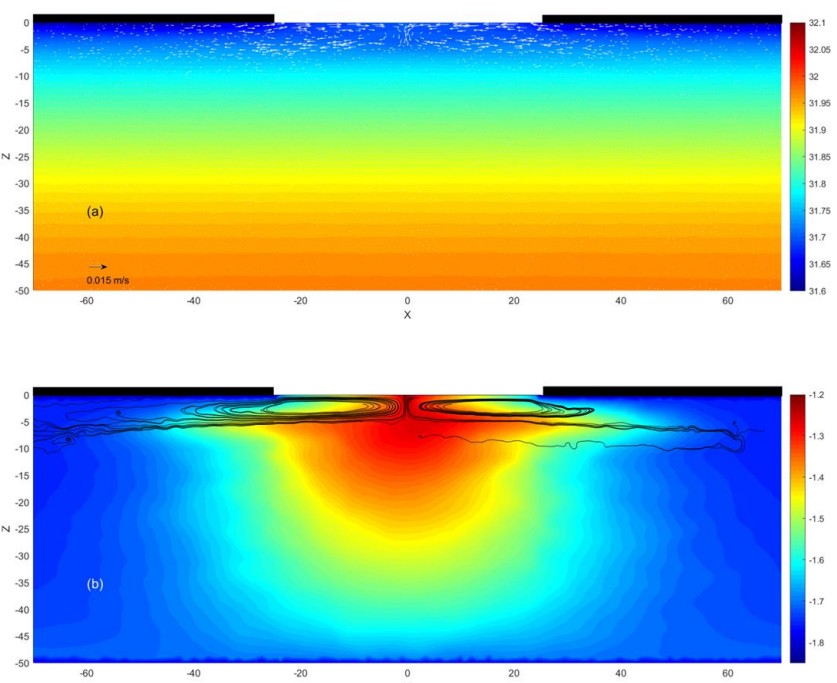

**Figure 4: (a) Salinity and current fields and (b) temperature distribution and streamlines with maximum incoming solar radiation**
**(label (d) in Figure 2). Resolution of the current field has been reduced to one third to facilitate visualization. Colour scale refers to PSU in (a) and ºC in (b).**

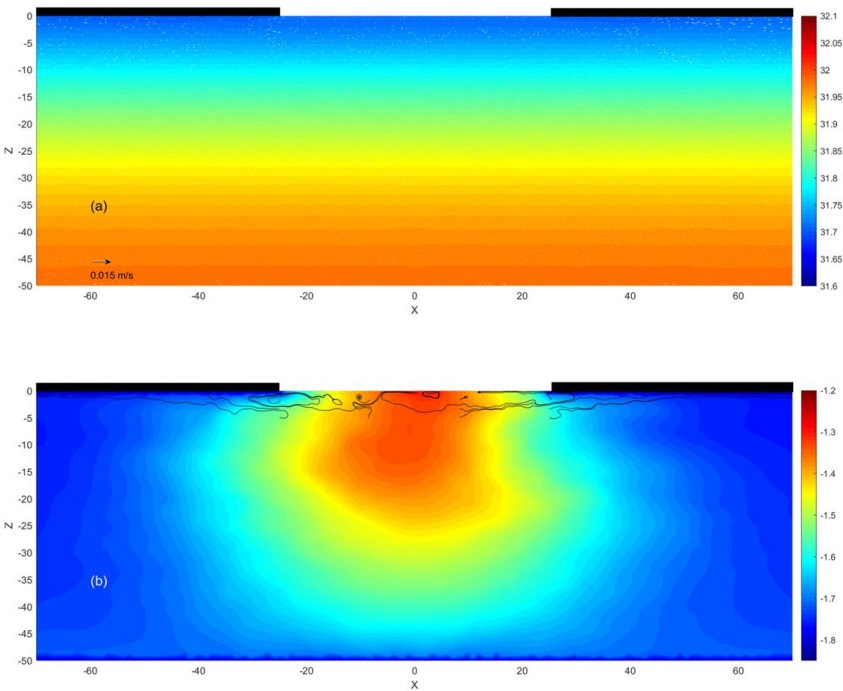

**Figure 5: Same as Figure 4 but for the situation with minimum solar radiation (label (a) in Figure 2).**




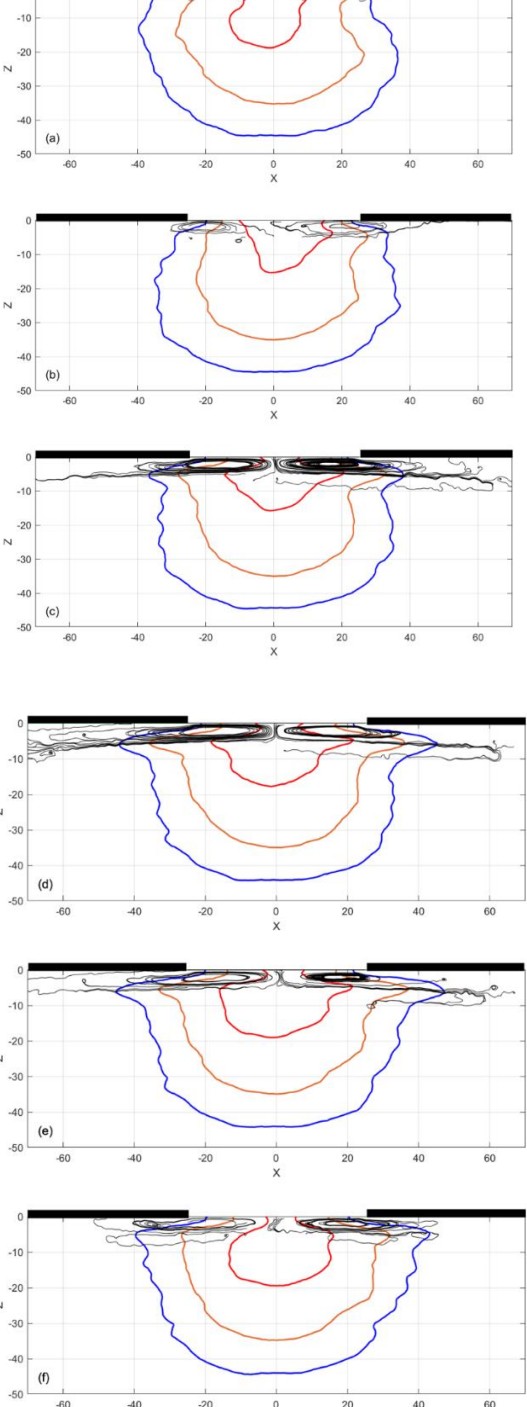

**Figure 6: Isothermals of temperature anomalies of 0 ºC (blue), 0.1 ºC (orange) and 0.25 ºC (red) and streamlines during different stages of incoming solar radiation. Figure alphabetic labels correspond to those displayed in Figure 2.**



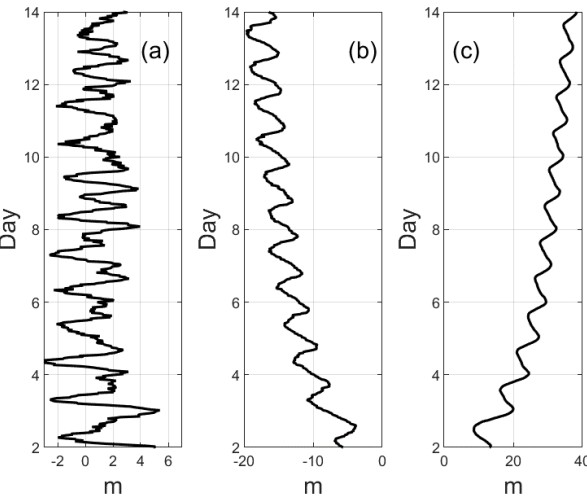

**Figure 7: Time evolution of the (a) bulk horizontal coordinate, (b) bulk vertical coordinate and (c) characteristic size of the isothermal corresponding to -1.5 ºC.**

Figure 6 shows that the subsurface core of warm waters undergoes through changes and distortions in shape, triggered by the variability of the convection cells. Shape modifications of the warm water core have been quantified by the time evolution of

the bulk horizontal and vertical positions of a particular temperature isothermal (-1.5 ºC) as well as by the characteristic size of the area it encloses. Oscillations of the bulk horizontal coordinate informs about the horizontal displacement and distortion of the core of warm water masses, Figure 7a. The power spectrum of the time series of horizontal displacements reveals that the reported variability is dominated by the inertial and daily frequencies. Daily oscillations are superimposed on a deepening trend of the bulk vertical coordinate, Figure 7b. The bulk vertical stretching occurs at a rate of -0.5 m per day. The growing of

the patch of warm waters is represented in Figure 7c. Similarly to the previous case, a daily oscillation is superimposed to a growing trend of 1 m per day of the representative length scale (an increase in area of 68 m2 per day). Thus, Figure 7 quantifies previous findings describing a core of warm waters growing and deepening while subjected to horizontal displacements and distortions.

The displacements of the thermal center of the water mass due to the convection cell dynamics have a signature in the heat

storage of the water column. In particular, the heat storage rate in the upper ocean is determined by (Moisan and Niiler, 1998),

$$\rho c_p \left( h \frac{\partial T_a}{\partial t} \right) \tag{16}$$

were $c_p$ is the specific heat of seawater, h =50 m is the depth of a chosen isotherm across which there is a minimal heat transfer

rate and from which the depth averaged temperature $T_a$ is calculated. In the present case, Eq (16) was calculated using hourly data with either centered, forward or backward differencing in time depending upon data availability.

The time evolution of the heat storage rate is characterized by the periodicity of a complicate pattern characterized by alternate positive and negative values, Figure 8. The dynamics of the heat storage rate is mainly dictated by the described variability of the convection cells. The heat storage rate of the states described in Figure 6 are marked in Figure 8 to facilitate comprehension.

A remarkable feature in the time variability of the heat storage rate is observed during the time evolution from stage (b) to (c). This time frame corresponds to the appearance and growing of the convection cells. In particular, Figure 8 evidences the surface propagation towards the lead center of the cold waters resulting from the sea ice melting. The two convection cells

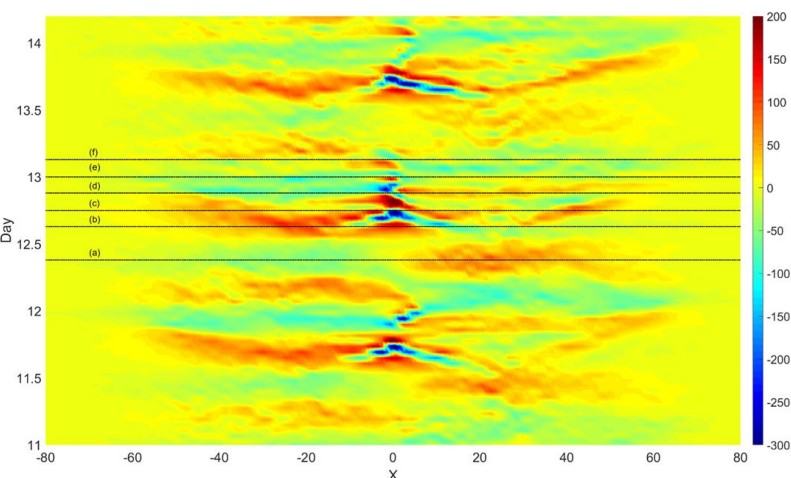

**Figure 8: Evolution of the heat storage rate. Alphabetic labels refer to those indicated in Figures 2 and 6. Colour scale refers to Wm⁻².**

reach simultaneously the lead center close to stage (c), generating the deepening of heated surface waters accumulated at this location by the circulation. The vertical convection of relatively warm waters results on an intense negative value heat storage rate at the lead center. Conversely, heat storage rate is positive at other lead locations due to the subsurface return flow of the heated waters. Heat storage gain prevails when evolving from stages (c) to (d), being the later the stage at maximum solar insolation. A significant part of the observed complex variability in the heat storage rate of the water column is related to changes in shape of the subsurface core of warm waters previously reported.

## 4 Discussion

The mixed layer circulation under a melted lead has been analysed in this study. The theoretical framework is defined in an idealized scenario that captures the fundamental physical aspects of the problem. These are defined by the incoming solar radiation, the sea ice melting and the lead geometry. Other forcing like longwave radiation, latent and sensible radiation heat fluxes, wind and surface waves that would exist in actual leads was not considered in this study. Paulson and Pegau (2001) reported net average values of -25, 5 and 3 Wm⁻² for the longwave radiation, sensible and latent heat fluxes, respectively, in a lead during summer. The sum of the net radiative, sensible and latent heat fluxes is small compared with the incoming shortwave radiation and reduced impact on the dynamics is assumed for the time scale considered. On the other hand, wind fetch is rather limited within the leads preventing the full development of the wave field (Pegau and Paulson, 2001). This is especially true in small leads. For this reason, the idealized scenario would be representative of leads under calm or small leads under gentle wind conditions during the summer season.

The proposed scenario has been mathematically formalized in terms of a nonhydrostatic, Boussinesq, rotating, two-dimensional model. This mathematical description is usual to simulate lead oceanography (Kozo, 1983; Smith and Morison, 1998) because it fully exploits the symmetry of the lead geometry to reduce the dimensionality of the problem. Its validity is constrained to the lead region far from boundaries and openings. A detailed simulation of the sea ice thermodynamics and dynamics is out of the scope of the present study. Instead, a methodology to provide reasonable boundary conditions for salinity fluxes from the melting ice cap is required. In this work, salinity fluxes resulting from sea ice melting have been parametrized in terms of an empirical relationship suggested by Perovich et al. (2011), relating the heat flux used in melting the bottom of the ice and the solar heat input to the ocean. A simple thermodynamic model translates the heat flux available for melting to





the required salinity flux boundary condition. A melting rate of 0.045 m d$^{-1}$ is obtained from the present approach for an ice

cap of 1 m thickness heated by the incoming solar radiation depicted in Figure 2. This value of the melting rate is reasonable close to the melting rate of 0.05 m d$^{-1}$ obtained with more sophisticated thermodynamic models of sea ice under similar conditions (Grumbine, 1994).

Numerical simulations of the idealized scenario reveal the daily generation of near surface convection cells. This circulation pattern results from the combined effect of the lead geometry, solar heating and sea ice melting. In the simulations, the

convection cells fade away when the incoming solar radiation is not enough to trigger a significant fresh water flux through ice melting. Convection rolls initiate to disappear when solar heating is below the threshold value of 100 W m$^{-2}$. No significant circulation pattern below the lead results during the insolation period below the threshold value.  While relatively constrained to near surface layer, the convection cells have revealed themselves an important mechanism to determine the thermodynamics of the mixed layer under the lead. In particular, convection cells routinely pump heated surface waters below the surface at the

lead center. This originates the existence of water masses below the lead, warmer than the surrounding environment. Warm water masses also spread out horizontally below the ice cap as a result of the circulation pattern. The patch of warm water masses stretches and deepens during the period of maximum solar radiation and contracts when the solar heating falls below the reported threshold. A net stretching and deepening results from the cycling process. Warm water masses are also subjected to asymmetric horizontal displacements/ distortions which origin is argued in terms of the asymmetry introduced by the

Coriolis term and cyclic forcing. The thermodynamic signature of this variability is captured in the heat storage rate of the water column due to its time differential nature.

A significant increase in the upper Arctic ocean heating has been observed in the last decades (Steele et al., 2008). Most of the oceanic heat is accumulated in the NSTM feature formed between mid-June and mid-July. Although the formation of the NSTM is still under debate, there is a scientific consensus in attributing the NSTM development mainly to the absorption of

solar radiation penetrating the upper ocean through the leads and melt ponds (Jackson et al., 2010). However, this mechanism does not exclude other contributing processes. Steele et al. (2011) found, using a Pan-Arctic circulation model, an additional NSTM formation process which origin is the surface cooling in the open waters. In addition, Gallaher et al. (2017) highlight from observations and a 1-D boundary layer model the relevance of season buoyancy and wind events to facilitate the development of the NSTM. These evidences seem to confirm that processes of different scale participate in the formation of

the NSTM feature. In this context, present results suggest a new local process that may contribute or facilitate the development of the NSTM in the leads. This is given by the convective transport of heat due to the circulation cells resulting from the sea ice melting and the particular lead geometry. This convective transfer would reinforce the heating of the subsurface layers by the direct absorption of solar radiation.

If predictions are accurate, global warming is expected to lead to a nearly ice-free Arctic Ocean in summer in a few decades.

This hypothesized scenario implies the substitution of the robust perennial ice cap by a weaker cover of seasonal young ice. This would facilitate the formation of melted leads at the beginning of the melting season, increasing their frequency in the ice cover. Thus, the effects of oceanographic processes at lead scale (including the one reported in this study) may enhance their global relevance. Of particular interest for this author, is that the expected future warming and deepening of the NSTM may result into a sound channel similar, but with different physical origin, to the "Beaufort lens" recently discovered in the Canadian

Arctic (Freitag et al., 2015). Assessing the mechanisms of formation of the NSTM feature, its physical properties, its resiliency along the year and its geographical distribution is of special relevance to determine the underwater acoustic landscape in the future Arctic Ocean.



**Competing interests**

The author declares that he has no conflict of interest.

**Acknowledgments**

This work has been funded by the project SAC000A06 of the NATO Allied Command Transformation-ACT. Comments to the manuscript from A. Russo are greatly appreciated.

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
