# Peer review of "A model for the Artic mixed layer circulation under a melted lead: Implications on the near-surface temperature maximum formation"

_The Cryosphere, 2020_

## Referee Comment (RC1) · Anonymous Referee #1 · 17 Feb 2021

This study uses a simple, idealized ocean model to argue for a new process contributing to the formation of the near-surface temperature maximum (a feature observed in the Western Arctic) relying on circulation associated with summer leads. My review is based on my experience with observations and climate-scale representations of sea ice and upper ocean processes. As such, I cannot comment directly on the suitability of the idealized numerical model.

Overall, this study presents an interesting new mechanism that could contribute formation of the NSTM locally in the Arctic. It is an appropriate subject for publication in The Cryosphere. However, the current manuscript leaves too many lingering questions

resulting from over-simplification of the problem and ignoring many important factors (i.e. wind, drift, appropriate solar forcing, sea ice geometry). This limits the utility of the study for informing understanding of physical processes in leads, and contribution to heat storage. I understand it is not likely that all realistic forcing conditions can be considered due to model constraints, but I think a revised version should at least demonstrate more consideration of which variables are important to include and discussion of the implications of those that remain excluded.

**Major comments:**

- Appropriate forcing. The SW forcing used here, shown in Figure 2 (from Mcphee Stanton, 1996) is for April north of Alaska. As the aim of this study is to explore circulation patterns of a summer lead with melting conditions, I see no reason why this would be the correct forcing to use. For the conditions you propose to explore, I would expect the forcing typical of Canada Basin (with minimal diurnal cycle) in July to be best suited – as this is where the NSTM is documented to form, but has more persistent summer ice pack than the area north of Alaska for which this forcing was obtained. Perhaps the forcing data from SHEBA would be appropriate.

- Consideration of geometry. Due to the scales over which the circulations are represented here, the fact that sea ice is not embedded in the ocean, but rather sitting atop (Figure 3) could have significant implications. It seems to me that this could disrupt the formation of circulation cells that are shown. If there is no way to represent this in the model, there needs to be significant discussion of the implications.

- Realism of atmospheric and ocean conditions. Ideally, some representation of impact of wind and/or sea ice drift can be included. (see e.g. Skyllingstad Paulson, 2005) Alternatively, more discussion of the extent to which this idealized scenario can/cannot represent realistic conditions is needed. There is also often

substantial shear between the ice and ocean during the summer in many parts of the ocean. How low would drift need to be to observe convection as simulated in these experiments? How gentle wind would have to be to be considered "wind-less" as in these experiments? How often do these conditions exist?

- Presentation of results. The connection of convection cells formed in the model with heat storage (i.e. in NSTM) could be shown more clearly. At current, it is hard to see that this mechanism could truly contribute to regional formation of NSTM. For example, Fig. 8 focuses on evolution of horizontal heterogeneity whereas my understanding is that NSTM formation will require vertical heterogeneity, as shown in Fig. 6. Consider how you can connect these results to what we typically expect to see with formation of NSTM (i.e. Steele et al., 2011).

**Additional comments**

- Title: "melted lead" seems a strange word choice – even during the summer they're still often dynamically formed. Perhaps 'summer lead' would be better.

- L37-53: It seems strange to focus the background material on winter lead processes, as the study here is nominally focused on summer leads. I would suggest shortening this discussion of leads in winter and expanding the introduction of leads in summer, as there is additional literature that provides important context not included here (e.g. Richter-Menge et al., 2001; Skyllingstad Paulson, 2005).

- L24-26: I believe this point is still disputed (see e.g.: Blackport Screen 2021, Screen et al., 2018 Nature Geoscience). Perhaps better to make a more vague statement that it likely influences Northern Hemisphere weather.

- L42-43: Please provide a citation showing observation/indication of this process

- L95: The albedo of open water is typically estimated at 0.067. Why do you use 0.02 here?

- L98-99: If the effects of melting on ice-plate geometry are not considered, the impacts of this should be indicated. Can you confirm that the freshwater flux does not exceed what is realistic based on the ice volume? What are the possible feedbacks associated with ice thinning? (I.e. reduction in drag, reduction in albedo, etc.)

- L169: It's wasn't very clear to me what initial temperature  salinity profiles were/why. I see now that they are shown in Fig. 1. How were these profiles chosen, and why is there an increase in salinity at 50 m? Please describe this in the text. Also, it would be better to start with a more realistic profile such as from observations (i.e. in Richter-Menge et al., or from ITPs)

- L220/Figure 4: Please show somewhere (or describe) the magnitude and time variation of the freshwater flux driving stratification and circulations.

- L225/Figure 6: It would be helpful to have on same figure the solar radiation values or time series and/or the times of each panel labeled.

- L225/Figure 6: What timeframe are the results being shown from? Are they from 1 specific date, or average from multiple days at the same time/solar radiation?

- L230/Figure 7: What are the bulk horizontal coordinate and bulk vertical coordinate? I assume these somehow correspond to the isothermals in Figure 6, but need to be defined.

- L273-274: More discussion on how you can ignore wind, drift etc is needed. Is there some other way you can parameterize the likely turbulent forcing in the near-surface? I suspect that the lack of background turbulence is leading to unrealistic horizontal heterogeneity (i.e. figures 4  5).

- L260/Fig. 8: This figure is the key result figure, and I find it does not adequately convey the information, or in the best way possible. Perhaps there is some more

simplified way of quantifying 'strength of circulation/cells' alongside vertical heat storage (i.e. integrated horizontally rather than vertically). The key feature of the NSTM is that it is sub-surface.

- Fig. 8: Also, consider re-orienting the time series (I feel time should progress downwards) and adding other supplementary information (i.e. gridlines for days, annotation of lead width, 0=white) that would make it easier to digest.

- L307-308: This statement is not very accurate. First of all, the NSTM is generally understood to be a regional feature (Canada Basin) so other features in the profile dominate elsewhere in the basin. Additionally, the NSTM captures the remnant warm water at the end of summer, but a large portion still goes to melt or back to the atmosphere. I have not seen a study quantifying the relative proportions to be able to say that a majority is captured in the NSTM.

---

## Referee Comment (RC2) · Anonymous Referee #2 · 23 Feb 2021

This paper presents an idealized simulation of ice-ocean interactions within a lead, with focus on the fluid dynamics below the ice. The study finds the daily generation of convection cells in response to shortwave-driven sea ice melt. I find the approach and results interesting and worthy of publication, but there are a number of issues related to the presentation of the study and the approach that I feel need to be addressed first.

Major

Introduction: -The introduction includes a very nice discussion of observational studies on leads, but what about other numerical studies? Ramudu et al., 2018 discusses NSTM specifically and might be a good starting point, but I imagine there are many

other numerical/fluid dynamics studies. -There is a very nice and thorough discussion of the seasonal evolution in a lead, but then the study focuses on the diurnal cycle. I suggest the author motivate this choice and perhaps focus the discussion on the time of year he is focusing on. -L68: The objective is stated here but it is very broad. Possible to be more specific on the question that is answered ? -It is mentioned in the methods that the ice will be assumed as motionless. This is okay but really needs to be stated up front that you are *only* considering thermodynamic drivers from sea ice melt/formation. This needs to be properly motivated in the introduction too.

Methodology: The NSTM seems like an important part of the research question (and is stated in the title). As far as I know, NSTM is mainly discussed in the context of the Canada Basin (and the references in the intro are all for the Canada Basin too). So why is the initial profile used from the Eurasian Basin? The Canada Basin is fresher, has a shallower mixed layer, and has a stronger halocline that is closer to the surface. This will likely change a lot. I strongly suggest the author consider downloading observations from a few ITPs in the Canada Basin, get a sense of what the halocline looks like there, and go from there. Alternatively, they could hunt for a representative profile in papers like Toole et al., 2010; Jackson et al., 2010; Timmermans 2015.

Results: -Validation: Once the model is spun up, are the T,S profiles still realistic? I think this needs to be included somewhere to know if we can trust the analysis that follows. -Sensitivity to parameters: How sensitive are the results to the choice of initial conditions and other parameters? One particularly interesting question that might be worth looking into is the sensitivity of the convection cells to the initial stratification (observations indicate that this is changing and having an impact on NSTM).

Minor

L39: heat flux from the ocean and the atmosphere? L40: although → even though? Despite? L45: Earlier studies that showed this: Morison 1978, Lemke 1983 L46: progress → progresses L46: to → and? L47: "Leads constitute then" ? L49: Add

comma after "In addition: L57: Maybe add Timmermans 2015 too L78: What is meant by lead spacing *below* 500m? Fig 1: What is meant by "ground values"? Fig 2: How do you choose what time periods to use (a-f), and what time of the year to use? L148-149: What is Qm exactly? It just says its "the linear relationship found by Perovich." But what is the equation? Fig. 6: Add legend for the colors. Please also add time of day/ heat flux in corner of each panel so we don't have to flip back to Figure 2. Fig. 7: How are the bulk coordinates and characteristic sizes defined? Average? Note: There appeared to be many grammatical errors/typos, I do not list them all here.

---

## Author Comment (AC1) · 23 Mar 2021

This study uses a simple, idealized ocean model to argue for a new process contributing to the formation of the near-surface temperature maximum (a feature observed in the Western Arctic) relying on circulation associated with summer leads. My review is based on my experience with observations and climate-scale representations of sea ice and upper ocean processes. As such, I cannot comment directly on the suitability of the idealized numerical model. Overall, this study presents an interesting new mechanism that could contribute formation of the NSTM locally in the Arctic. It is an appropriate subject for publication in The Cryosphere. However, the current manuscript

leaves too many lingering questions resulting from over-simplification of the problem and ignoring many important factors (i.e. wind, drift, appropriate solar forcing, sea ice geometry). This limits the utility of the study for informing understanding of physical processes in leads, and contribution to heat storage. I understand it is not likely that all realistic forcing conditions can be considered due to model constraints, but I think a revised version should at least demonstrate more consideration of which variables are important to include and discussion of the implications of those that remain excluded.

R: I thank the reviewer for his/her useful comments that will certainly improve the current manuscript. This study investigates the circulation resulting from the combined effect of lead geometry, solar radiation and sea ice melting. Thus, only geometry and thermodynamic drivers are considered in the study. Results indicate that this basic configuration develops a circulation pattern that may have implications in the temperature distribution in the Arctic mixed layer. The basic circulation pattern could be modulated or inhibited by other forcings, not considered in the current scenario. Hopefully, this work will motivate researchers to investigate the resilence of this circulation pattern due to the effects of other forcings and evaluate its relevance in a more complex and realistic scenarios. I agree with the reviewer that the manuscript would benefit of a deeper discussion along this line. The following modifications to the current version will be implemented according to the referee's comments.

Major comments: - Appropriate forcing. The SW forcing used here, shown in Figure 2 (from Mcphee Stanton, 1996) is for April north of Alaska. As the aim of this study is to explore circulation patterns of a summer lead with melting conditions, I see no reason why this would be the correct forcing to use. For the conditions you propose to explore, I would expect the forcing typical of Canada Basin (with minimal diurnal cycle) in July to be best suited – as this is where the NSTM is documented to form, but has more persistent summer ice pack than the area north of Alaska for which this forcing was obtained. Perhaps the forcing data from SHEBA would be appropriate.

R: Due to geographical proximity and the capability to perform future observations, the

author is particularly interested in the Eurasian basin, where the NSTM and its impact on sound propagation have already been reported (Carmack et al., 2015, Freitag et al., 2015). Even with the described level of abstraction, I fully agree with the reviewer that a more consistent scenario can be accomplished in the study. In particular, new simulations will be done with a representative incoming shortwave radiation during June in the Arctic Eurasian Basin. This is more consistent with the schematic stratification of the model, which is typical of the Eurasian Basin, and with the solar forcing during the melting season.

- Consideration of geometry. Due to the scales over which the circulations are represented here, the fact that sea ice is not embedded in the ocean, but rather sitting atop (Figure 3) could have significant implications. It seems to me that this could disrupt the formation of circulation cells that are shown. If there is no way to represent this in the model, there needs to be significant discussion of the implications.

R: The representation of the sea ice cap follows the same geometrical description found in previous numerical works considering leads at similar scales (Kozo, 1983; Smith and Morison, 1993; Smith and Morison, 1998). The proposed idealization misrepresents the effect of the melting on the lateral edge of the sea ice cap, which contribution is negligible in the current case. No significant circulation effects would be expected with this refinement in the boundary condition in a circulation pattern resulting from buoyancy. Smith IV D. and Morison J., Nonhydrostatic haline convection under leads in sea ice. J. Geophys. Res., 103, 3233-3247, doi: 10.1029/97JC02262, 1998. Smith, D.C., IV, and J. H. Morison, A numerical study of haline convection beneath leads in sea ice, J. Geophys. Res., 98, 10,069- 10,087, 1993. Kozo, T. L., Initial model results for Arctic mixed layer circulation beneath a refreezing lead, J. Geophys. Res., 88, 2926-2934, 1983. Lateral lead effects were evaluated by Skyllingstad et al. (2005) in a very different framework (see below).

- Realism of atmospheric and ocean conditions. Ideally, some representation of impact of wind and/or sea ice drift can be included. (see e.g. Skyllingstad Paulson, 2005)

Alternatively, more discussion of the extent to which this idealized scenario can/cannot represent realistic conditions is needed. There is also often substantial shear between the ice and ocean during the summer in many parts of the ocean. How low would drift need to be to observe convection as simulated in these experiments? How gentle wind would have to be to be considered "wind-less" as in these experiments? How often do these conditions exist?

R: It is now clarified in the introduction that: An axisymmetric geometry and particular thermodynamic forcings are common features to summer leads. For this reason, this study focusses on the circulation under a summer lead resulting from the combined effect of the lead geometry, solar radiation and sea ice melting as well as its effect, if any, in the heat exchanges through the lead. The ultimate objective of the study is to assess if the circulation pattern under the lead could contribute to the formation of the NSTM.

The impact of wind is discussed in the manuscript. In particular, it is mentioned: On the other hand, wind fetch is rather limited within the leads preventing the full development of the wave field (Pegau and Paulson, 2001). This is especially true in small leads. For this reason, the idealized scenario would be representative of leads under calm or small leads under gentle wind conditions during the summer season.

The present study and Skyllingstad's significantly differ in the scope and physical/numerical settings. In particular, Skyllingstad et al. (2005) attempts to provide insight on how leads regulate lateral melting by trapping fresh water under wind dominated conditions. Their physical and numerical setup focusses only in the first 4m of the surface layer and on horizontal scales of maximum 26m. Their simulations were of relatively short duration (4 hours). Instead, this study investigates the circulation resulting in the Arctic mixed layer under a lead due to geometry and thermodynamic drivers. Thus, the physical/numerical setting of this study considers vertical scales of 50 m and horizontal scales of hundreds of meters. In addition, simulations cover time scales of weeks, instead few hours. Melting involves the whole ice cap and not only the

lateral melting at the lead edge. The difference in scope, settings and methodologies makes difficult a comparison between both works, as we are looking at different physical phenomena. Finally, notice that Skyllingstad et al (2005) also considers motionless sea ice as reported in Section 2.2. Initial Conditions and Simulation Parameters, indicate that "...so we did not consider cases with ice motion. Ice motion with very large leads (âĹij500 m) might have a greater influence, however, here our focus was on small leads." It will be specified in the manuscript that: The ice plates are assumed motionless as only thermodynamic drivers are considered in this study. The new version of the manuscript will further discuss the limitations of the idealized scenario.

- Presentation of results. The connection of convection cells formed in the model with heat storage (i.e. in NSTM) could be shown more clearly. At current, it is hard to see that this mechanism could truly contribute to regional formation of NSTM. For example, Fig. 8 focuses on evolution of horizontal heterogeneity whereas my understanding is that NSTM formation will require vertical heterogeneity, as shown in Fig. 6. Consider how you can connect these results to what we typically expect to see with formation of NSTM (i.e. Steele et al., 2011).

R: The heat storage rate involves, by definition, the depth averaged temperature (Eq 16). The Figure attempts to show the effect of the circulation cells on this magnitude. As it is described, the cells introduce alternate patterns of negative and positive heat storage rate correlated with the forcing and the subsurface injection of heat due to the development of the cells. This correlation is indicated by the alphabetic labels referred to those indicated in Figures 2 and 6 (see caption of Figure 8). Notice that Fig. 6 and 7 already provide information about the bulk heat storage by tracking a determine isotherm. Figure 8 completes this information showing the evolution of the heat storage rate. The existing explanation will be further clarified, the relationship with Figures 2 and 6 reinforced and new figures considered. Steele et al. (2011) suggested, using a large scale Pan-Arctic circulation model, an additional mechanism for NSTM formation process which origin is the surface cooling and the subsurface absorption of solar

radiation in the open waters. Instead, the scientific consensus attributes the NSTM development mainly to the absorption of solar radiation penetrating the upper ocean through the leads and melt ponds (Jackson et al., 2010). This work contributes and complements the latter hypothesis, being conceptually far from the formation mechanism proposed by Steele at al. (2011). As it is mentioned in the discussion, these studies (Jackson et al., 2010; Steele et al. 2011; Gallaher et al. 2017) suggest that different mechanisms can generate similar NSTM layers.

Additional comments - Title: "melted lead" seems a strange word choice – even during the summer they're still often dynamically formed. Perhaps 'summer lead' would be better.

R: The title will be changed as indicated

- L37-53: It seems strange to focus the background material on winter lead processes, as the study here is nominally focused on summer leads. I would suggest shortening this discussion of leads in winter and expanding the introduction of leads in summer, as there is additional literature that provides important context not included here (e.g. Richter-Menge et al., 2001; Skyllingstad Paulson, 2005).

R: The introduction will be revised accordingly to the comment and the references incorporated

- L24-26: I believe this point is still disputed (see e.g.: Blackport Screen 2021, Screen et al., 2018 Nature Geoscience). Perhaps better to make a more vague statement that it likely influences Northern Hemisphere weather.

R: The text will be modified according to the comment

- L42-43: Please provide a citation showing observation/indication of this process

R: The following references are added: Morison J. H., McPhee, M. G., Lead convection measured with an autonomous underwater vehicle, J. Geophys. Res., 103, 3257-3281, 1998. https://doi.org/10.1029/97JC02264 Morison, J. H., McPhee, M. G., Curtin, T. B.,

Paulson, C. A., The oceanography of winter leads, J. Geophys. Res., 97, 11199-11218, 1992. https://doi.org/10.1029/92JC00684

- L95: The albedo of open water is typically estimated at 0.067. Why do you use 0.02 here?

R: Li et al, 2006 provide different models of the ocean surface albedo (Cox and Munk, 1954; Hansen 1983; Preisendorfer and Mobley, 1986; Hansen et al., 1983). At solar zenith and low wind speeds the ocean surface albedo is 0.02 in all models.

- L98-99: If the effects of melting on ice-plate geometry are not considered, the impacts of this should be indicated. Can you confirm that the freshwater flux does not exceed what is realistic based on the ice volume? What are the possible feedbacks associated with ice thinning? (I.e. reduction in drag, reduction in albedo, etc.)

R: The new version will indicate that: the model applies to large enough sea ice plates and short enough times to hold constant ice properties (e.g., geometry, mass, salinity). It is mentioned in the discussion: A melting rate of 0.045 m d-1 is obtained from the present approach for an ice cap of 1 m thickness heated by the incoming solar radiation depicted in Figure 2. This value of the melting rate is reasonable close to the melting rate of 0.05 m d-1 obtained with more sophisticated thermodynamic models of sea ice under similar conditions (Grumbine, 1994). Full dedicated studies would be required to address the level of knowledge granularity posed in the last question.

- L169: It's wasn't very clear to me what initial temperature salinity profiles were/why. I see now that they are shown in Fig. 1. How were these profiles chosen, and why is there an increase in salinity at 50 m? Please describe this in the text. Also, it would be better to start with a more realistic profile such as from observations (i.e. in Richter-Menge et al., or from ITPs)

R: The selected profiles follow the schematic temperature and salinity vertical structure in the upper 500m of the Eurasian Basin in the Arctic Ocean as described in Davis et al

(2016). According to the schematic structure, the model considers constant profiles of temperature and salinity in the mixed layer. Characteristic values were obtained from Davies et al., 2016. The increase of salinity at 50 m depth represents the end of the mixed layer and the beginning of the pycnocline. The information of the profile structure below the mixed layer is used to fix Dirichlet and Neumman boundary conditions for temperature and salinity, respectively, at the bottom of the mixed layer. In other words, the model does not consider temperature and salinity profiles below this depth, but adequate boundary conditions.

- L220/Figure 4: Please show somewhere (or describe) the magnitude and time variation of the freshwater flux driving stratification and circulations.

R: The information will be included in the Result section

- L225/Figure 6: It would be helpful to have on same figure the solar radiation values or time series and/or the times of each panel labeled.

R: The information is provided with the alphabetic labels which corresponds to those displayed in Figure 2, as indicated in the Figure caption. I agree that adding the value of the incoming shortwave radiation would further clarify the link.

- L225/Figure 6: What timeframe are the results being shown from? Are they from 1 specific date, or average from multiple days at the same time/solar radiation?

R: It is written: The particular analysis during days 12 and 13 follows as representative of the oceanographic conditions obtained after the model spin up. This point will be further clarified.

- L230/Figure 7: What are the bulk horizontal coordinate and bulk vertical coordinate? I assume these somehow correspond to the isothermals in Figure 6, but need to be defined.

R: Yes, it is mentioned in line 234-236: Shape modifications of the warm water core have been quantified by the time evolution of the bulk horizontal and vertical positions

of a particular temperature isothermal (-1.5 oC) as well as by the characteristic size of the area it encloses.

- L273-274: More discussion on how you can ignore wind, drift etc is needed. Is there some other way you can parameterize the likely turbulent forcing in the near-surface? I suspect that the lack of background turbulence is leading to unrealistic horizontal heterogeneity (i.e. figures 4, 5).

R: The aspects related to wind and drift were previously answered. Wind is not considered in the present study. Instead, the study focuses on the circulation resulting from the lead geometry, solar radiation and ice melting. The model uses a Large Eddy Simulation (LES) scheme to parametrize the turbulence levels at all depths (Smagorinsky 1963, 1993). Turbulence levels range from 5 10-3 m2/s and 10-5 m2/s (i.e. from thousand to ten times higher than the molecular viscosity) in agreement with eddy activity expected from the resulting current intensity. The flow is dominated by convection under the prescribed calm conditions. The mentioned horizontal heterogeneity is induced by the cells. Results agree with expectations according to the physics and prescribed conditions. Temperature heterogeneities due to convection-dominated flows are common in ocean flows at different scales.

- L260/Fig. 8: This figure is the key result figure, and I find it does not adequately convey the information, or in the best way possible. Perhaps there is some more simplified way of quantifying 'strength of circulation/cells' alongside vertical heat storage (i.e. integrated horizontally rather than vertically). The key feature of the NSTM is that it is sub-surface.

R: This point was previously discussed. Notice that the figure refers to the heat storage rate which is, by definition, a vertical averaged magnitude. The vertical signature of the NSTM was shown in previous figures. Figure 8 displays the signature of the time variability of the cells on the heat storage rate.

- Fig. 8: Also, consider re-orienting the time series (I feel time should progress downwards) and adding other supplementary information (i.e. gridlines for days, annotation of lead width, 0=white) that would make it easier to digest.

R:The figure will be re-oriented

- L307-308: This statement is not very accurate. First of all, the NSTM is generally understood to be a regional feature (Canada Basin) so other features in the profile dominate elsewhere in the basin. Additionally, the NSTM captures the remnant warm water at the end of summer, but a large portion still goes to melt or back to the atmosphere. I have not seen a study quantifying the relative proportions to be able to say that a majority is captured in the NSTM.

R:The sentence will be rewritten. Please, notice that the NSTM was also reported in the Arctic Eurasian Basin (Carmack et al., 2015, Freitag et al., 2015).

---

## Author Comment (AC2) · 23 Mar 2021

This paper presents an idealized simulation of ice-ocean interactions within a lead, with focus on the fluid dynamics below the ice. The study finds the daily generation of convection cells in response to shortwave-driven sea ice melt. I find the approach and results interesting and worthy of publication, but there are a number of issues related to the presentation of the study and the approach that I feel need to be addressed first.

R: I thank the reviewer for his/her encouraging comments to the manuscript. His/Her suggestions have been considered to improve the manuscript as detailed below:
Major - Introduction: -The introduction includes a very nice discussion of observational studies on leads, but what about other numerical studies? Ramudu et al., 2018 discusses NSTM specifically and might be a good starting point, but I imagine there are many other numerical/fluid dynamics studies. -There is a very nice and thorough discussion of the seasonal evolution in a lead, but then the study focuses on the diurnal cycle. I suggest the author motivate this choice and perhaps focus the discussion on the time of year he is focusing on.

R: I thank the reviewer for highlighting this reference. I agree with the reviewer that the Introduction Section lacks of an appropriate description about numerical studies related to the subject investigated. A paragraph will be included in this Section summarizing the numerical findings, in particular those of Ramudu et al., 2017.

-L68: The objective is stated here but it is very broad. Possible to be more specific on the question that is answered?

R: It is proposed to rewrite the sentence: An axisymmetric geometry and particular thermodynamic forcings are common features to summer leads. For this reason, this study focusses on the circulation under a summer lead resulting from the combined effect of the lead geometry, solar radiation and sea ice melting as well as its effect, if any, in the heat exchanges through the lead. The ultimate objective of the study is to assess if the circulation pattern under the lead could contribute to the formation of the NSTM.

-It is mentioned in the methods that the ice will be assumed as motionless. This is okay but really needs to be stated up front that you are *only* considering thermodynamic drivers from sea ice melt/formation. This needs to be properly motivated in the introduction too.

R: In addition to the answer provided above, it is proposed to rewrite the sentence: The ice plates are assumed motionless as only thermodynamic drivers are considered in this study.

-Methodology: The NSTM seems like an important part of the research question (and is stated in the title). As far as I know, NSTM is mainly discussed in the context of the Canada Basin (and the references in the intro are all for the Canada Basin too). So why is the initial profile used from the Eurasian Basin? The Canada Basin is fresher, has a shallower mixed layer, and has a stronger halocline that is closer to the surface. This will likely change a lot. I strongly suggest the author consider downloading observations from a few ITPs in the Canada Basin, get a sense of what the halocline looks like there, and go from there. Alternatively, they could hunt for a representative profile in papers like Toole et al., 2010; Jackson et al., 2010; Timmermans 2015.

R: Due to geographical proximity and the capability to perform future observations, the author is particularly interested in the Eurasian basin where the NSTM layer and its impact on sound propagation have been reported (Carmack et al., 2015, Freitag et al., 2015). The NSTM layer in the Eurasian Basin generates a sound channel (30-150 m depth) wider and deeper than in the Canadian Basin ($\sim$15-50m). The future warming and deepening of the NSTM resulting from climate projections in the Arctic Eurasian Basin, would drastically modify the underwater soundscape in the region. For greater geographical and seasonal consistency with the stratification employed in the model, new simulations will be done with a representative incoming shortwave radiation during June in the Eurasian Basin.

-Results: -Validation: Once the model is spun up, are the T,S profiles still realistic? I think this needs to be included somewhere to know if we can trust the analysis that follows.

R: This analysis will be included in the Result Section of the new version.

-Sensitivity to parameters: How sensitive are the results to the choice of initial conditions and other parameters? One particularly interesting question that might be worth looking into is the sensitivity of the convection cells to the initial stratification (observations indicate that this is changing and having an impact on NSTM).

R: Due to the long simulation runs (about 10 days each) only a limited sensitivity test can be done. In particular, it will be considered the sensitivity of the circulation cells with a shallower pycnocline.

Minor -L39: heat flux from the ocean and the atmosphere?

R: OK

-L40: although → even though? Despite?

R: OK

-L45: Earlier studies that showed this: Morison 1978, Lemke 1983

R: The following references are added: Lemke, P., Manley, T.O., The seasonal variation of the mixed layer and the pycnocline under polar sea ice, J. Geophys. Res., 89, 6494-6504, 1984. https://doi.org/10.1029/JC089iC04p06494 Morison, J., Smith, J. D., Seasonal variations in the upper Arctic Ocean at observed at T-3. J. Geophys. Res., 8, 753-756, 1981. https://doi.org/10.1029/GL008i007p00753

-L46: progress → progresses

R: OK

-L46: to → and?

R: OK

- L47: "Leads constitute then" ?

R: OK L49: Add comma after "In addition:

R: OK

-L57: Maybe add Timmermans 2015 too

R: Added: M. L. Timmermans, The impact of stored solar heat on Arctic sea ice growth, Geophys. Res. Lett., 6399-6406, 2015. http://doi.org/10.1002/2015GL064541

-L78: What is meant by lead spacing \*below\* 500m?

R: Corrected to: smaller than

-Fig 1: What is meant by "ground values"?

R: Corrected to initial temperature and salinity values

-Fig 2: How do you choose what time periods to use (a-f), and what time of the year to use?

R: The time periods were selected to be representative of the variation of the forcing by evenly distributing some time stations along the daily forcing evolution (Figure 2) to characterize the circulation patterns at this time stations. The forcing corresponds to the Julian day 99 of 1992. This would correspond to April. New simulations are done with an incoming short radiation forcing from June.

-L148-149: What is Qm exactly? It just says its "the linear relationship found by Perovich." But what is the equation?

R: It is a linear relationship between the heat used in bottom melt (y) and the solar heat input to the ocean (x) with slope 0.89 (y$\sim$0.89x-57). Perovich et al. 2011, indicate that: "The slope of the line is 0.89, indicating an almost one-to-one increase in bottom melting with solar heat input to the ocean. The relationship holds for observations that vary widely in geographic location, ice concentration and bottom melting. This argues that the primary source of heat for bottom melting of the ice is solar radiation absorbed in areas of open water (Maykut and McPhee, 1995; Perovich and others, 2008)."

-Fig. 6: Add legend for the colors. Please also add time of day/ heat flux in corner of each panel so we don't have to flip back to Figure 2.

R: This will be included in the figures

-Fig. 7: How are the bulk coordinates and characteristic sizes defined? Average? Note: There appeared to be many grammatical errors/typos, I do not list them all here.

R: Yes, average values. Typos will be corrected in the new version